# Learning Modulated Transformation in GANs

**Ceyuan Yang[1]    Qihang Zhang[2]    Yinghao Xu[2]    Jiapeng Zhu[3]    Yujun Shen[4]    Bo Dai[1]**

[1]Shanghai AI Laboratory    [2]CUHK    [3]HKUST    [4]Ant Group

## Abstract

The success of style-based generators largely benefits from style modulation, which helps take care of the cross-instance variation within data. However, the instance-wise stochasticity is typically introduced via regular convolution, where kernels interact with features at some *fixed* locations, limiting its capacity for modeling geometric variation. To alleviate this problem, we equip the generator in generative adversarial networks (GANs) with a plug-and-play module, termed as modulated transformation module (MTM). This module predicts spatial offsets under the control of latent codes, based on which the convolution operation can be applied at *variable* locations for different instances, and hence offers the model an additional degree of freedom to handle geometry deformation. Extensive experiments suggest that our approach can be faithfully generalized to various generative tasks, including image generation, 3D-aware image synthesis, and video generation, and get compatible with state-of-the-art frameworks without any hyper-parameter tuning. It is noteworthy that, towards human generation on the challenging TaiChi dataset, we improve the FID of StyleGAN3 from 21.36 to 13.60, demonstrating the efficacy of learning modulated geometry transformation. Code and models are available at https://github.com/limbo0000/mtm.

## 1   Introduction

Generative models are distinguished from other vision models by creating new data. The crux of learning a generative model is to mimic the observed data distribution through modeling as many data variations as possible. For example, a good generative model learned on a collection of cat images is expected to produce cats with diverse breeds, postures, hair colors, *etc.* Among all kinds of generative models [27, 43, 18, 48, 47], generative adversarial networks (GAN) [13] have received wide attention thanks to their flexibility and fast inference speed.

The recent success of GANs lies in the design of style-based generator [23]. Inspired by adaptive instance normalization (AdaIN) [19], StyleGAN [23] introduces layer-wise style codes to modulate the per-layer feature map from the generator. As these style codes are instance-specific, the stochasticity manifests itself in different ways of feature modulation, which finally contributes to diverse generation. Nevertheless, it is observed that GANs usually achieve good performance on single-object and aligned datasets, where all instances are with similar geometry, yet struggle in handling complex data distribution, such as ImageNet [10] that consists of various categories of objects, or TaiChi players [41] that make irregular movements.

We argue that such an issue may get caused by style modulation failing to model large geometry variations within data. Concretely, the operation of AdaIN equally acts on the feature map at every spatial location, leaving the problem of geometry modeling to the convolution operation together with its receptive field. Unfortunately, a regular convolution samples the input feature map at *fixed* locations, as shown in Fig. 1a. In this way, the model applies a fixed receptive field for all instances, limiting its capacity for learning geometry variations.

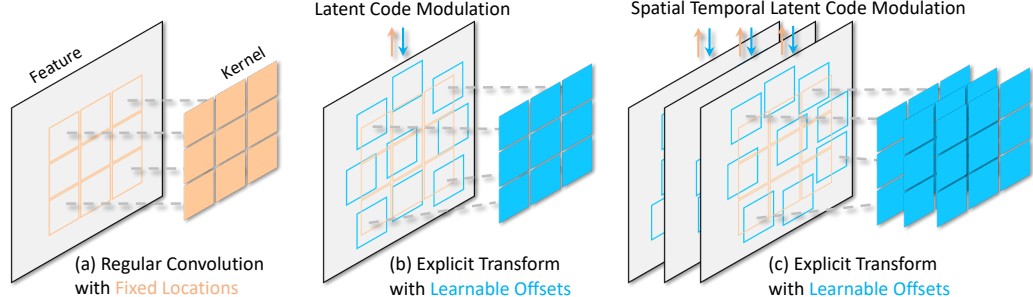

Figure 1: **Illustration of our** MTM. (a) Regular convolution interacts with features at *fixed locations*. (b) Our module predicts the offsets under the latent code modulation, based on which convolution can operate at *variable locations* for different instances. (c) When generating videos, the offsets with respect to each frame is jointly modulated by the spatial and temporal latent codes. With these learnable offsets, this module offers an additional degree of freedom to handle geometry transformation.

To tackle this obstacle, we propose modulated transformation module (MTM) that allows the convolutional kernels to interact input features at *variable* locations, as shown in Fig. 1b. In particular, given the feature map from the previous stage, we first predict an offset for each spatial point and then use the predicted offsets to warp the input feature before performing convolution. Here, the offset prediction is conditioned on the latent code and hence varies across instances. That way, the model is supported with an additional degree of freedom to learn geometry variation explicitly. It is noteworthy that, our MTM is also applicable to video generation, as shown in Fig. 1c, to help model the geometry transformation over time.

We evaluate our approach on a range of generative tasks, including image generation, 3D-aware image synthesis, and video generation. For each task, we choose the state-of-the art frameworks, *i.e.*, StyleGAN2 [24], StyleGAN3 [26], EG3D [7], and StyleSV [61] as our strong baselines. Extensive results on ImageNet [10], LSUN [58], YouTube Driving [60], SkyTimelapse [53], and TaiChi [41] suggest that MTM is capable of consistently boosting the performance, demonstrating its efficacy and generalizability. More importantly, MTM works as a plug-and-play module such that it can be effortlessly deployed to popular GAN-based frameworks even without any hyper-parameter tuning. Hopefully, our efficient MTM could serve as a basic operation in the future design of generative models.

## 2   Related work

**Image generation with GANs.**  Generative adversarial network (GAN) [13] has significantly advanced image generation. Typically, a two-player game is played between a generator and a discriminator. The generator aims to reproduce the real data distribution, while the discriminator aims to distinguish fake samples generated by the generator from the real ones. Many efforts have been made to design better architectures for both generators and discriminators [34, 15, 4, 22, 23, 38, 39, 21], to regularize discriminators for training stability [1, 15, 30, 57, 29, 2], and to bring various inductive biases [32, 6, 14, 54, 40, 7, 62, 55, 33, 50, 16, 63] to support explicit control. With the rapid development of generative models, a family of the style-based generators [23] stands out due to their photo-realistic synthesis and editing applications. In this work, we start with this popular generator and introduce a new module that targets at explicitly modeling the large geometric variation, leading to a better synthesis quality on various benchmarks.

**Video generation with GANs.**  Many attempts have been made to leverage GANs to synthesize videos from latent space. Video generation [44, 36, 45, 11, 42, 59, 49, 37, 8, 52, 61, 5] also grows up by incorporating the latest designs of image generators with temporal modeling. For example, TGAN [36] and MoCoGAN [45] explicitly disentangle the motion from the content, which is widely adopted in many years. Considering the dominant advantage of the style-based generator in image synthesis, multiple successful designs are further developed into video generation [42, 5, 61]. In this

work, we generalize the proposed module to video generation and demonstrate that it could also lead to consistent gains of synthesized video quality.

**Spatial transformation in neural networks.** In the field of visual perception, considering the inherent limitation of regular convolutions, spatial transformer network (STN) [20] first proposed to warp the whole intermediate representations according to learned global affine transformation. Deformable ConvNets (DCN) [9] further leveraged learnable offsets to sample intermediate features in a local and dense manner. Therefore, the spatial representations could be arbitrarily rearranged to represent the large-scale geometric variation. Its second version DCNv2 [64] enhanced the capacity and simplified the training strategy. The latest version DCNv3 [51] has been a vital component for the large-scale vision foundation models. Our work is very closed to the family of these spatial transformation/deformation networks. Differently, the predicted offsets are modulated by the latent code which does not exist in prior perception tasks. Moreover, to the best of our knowledge, it is the first time to introduce such explicit transformation into visual generation tasks.

## 3 Method

To facilitate generative adversarial networks in synthesizing data with large geometric transformations such as unaligned ImageNet images and TaiChi videos, we propose the modulated transformation module (MTM) that could explicitly deform the spatial geometry of given intermediate representations. Sec. 3.1 first presents how a regular convolution is performed in detail and shows a simple yet effective improvement that could handle large geometric variation. Sec. 3.2 further illustrates how to incorporate such a module into prior generative models for content generation (*i.e.*, images and videos). At last, Sec. 3.3 describes the training details including learning objectives and implementations.

### 3.1 Explicit spatial transformations

**Regular convolution.** Regular convolutional operations are inherently limited to model large geometric transformation/deformations, originating from their fixed shape of kernels. Specifically, given the intermediate representation $\mathbf{x}$ and the corresponding output $\mathbf{y}$ after a regular convolution, we let $\mathbf{x}(\mathbf{p})$ and $\mathbf{y}(\mathbf{p})$ denote the feature at location $\mathbf{p}$ respectively. Taking a convolution with a $3 \times 3$ kernel as an example, the regular convolution could be applied by sampling features at 9 locations:

$$\mathbf{y}(\mathbf{p}) = \sum_{i=1}^{9} w_i \cdot \mathbf{x}(\mathbf{p} + p_i), \tag{1}$$

where $w_i$ is the $i-$th weight of the convoultional kernel and $p_i \in [-1, 0, 1]^\wedge 2$ denotes the pre-defined spatial offset. In particular, current GANs usually adopt the modulated convolutions [23] whose weights would be modulated by the latent codes. Here the modulation is omitted for brevity. Such an operation aggregates the local information of a rectangular window (*e.g.*, $3 \times 3$) centered at location $\mathbf{p}$. For visual concepts whose information is scattered at irregular locations, it's thus hard for generative adversarial networks with regular convolutions to handle them effectively. Although ideally this limitation can be alleviated by stacking several layers and accumulating a sufficiently large receptive field, we empirically found in practice such a bypass struggles to draw non-rigid visual concepts. We therefore introduce the learnable offsets to GANs and enhance their ability with negligible computational overhead.

**Geometric transformations with learnable offsets.** Akin to the family of spatial transformer networks [20, 9, 64, 51], we introduce the learnable offsets $\Delta\mathbf{p}$ that explicitly rearrange the spatial locations of intermediate representations first. The regular convolution could be modified as

$$\mathbf{y}(\mathbf{p}) = \sum_{i=1}^{9} w_i \cdot \mathbf{x}(\mathbf{p} + p_i + \Delta p_i). \tag{2}$$

Here $\Delta p_i$ stands for the learnable offset for the $i-$th location. In particular, representation at location $\mathbf{p} + p_i + \Delta p_i$ would be obtained through the bilinear interpolation since the final offset would be fractional. Accordingly, the original fixed shape of the regular convolution would be arbitrarily reshaped to enable a larger receptive field, facilitating the geometric variation modeling simultaneously. To make the additional offsets data-dependent, prior literature regarding visual

perception [20, 9, 64, 51] usually predict offsets through an extra convolution (`Conv`) as

$$\Delta \mathbf{p} = \texttt{Conv}(\mathbf{x}). \tag{3}$$

Thus, for each location of representation $\mathbf{x}$, 18 scalars are predicted to form 9 pairs of spatial offsets. This could be easily implemented by setting the number of output dimensions as 18 regarding the extra convolution.

**Latent code modulation.** Different from perception tasks that often take diverse images as inputs, generation tends to produce realistic images from latent codes $\mathbf{z}$ that are randomly sampled from a pre-defined distribution (*e.g.*, normal distribution). Therefore, we further leverage the latent code and improve the offset prediction through the latent code modulation:

$$\Delta \mathbf{p} = \texttt{ModConv}(\mathbf{x}, \mathbf{z}). \tag{4}$$

Here, `ModConv` denotes the modulated convolution proposed by Karras *et. al.* [24]. Concretely, `ModConv` allows the input latent code $\mathbf{z}$ to modify the weight $w$ of convolutions first by multiplication (*i.e.*, $w' = w \cdot \mathbf{z}$). Such an improvement further makes the predicted offsets much more instance-specific since the corresponding latent code indeed contains all semantics of the synthesized output. Meanwhile, the optimization of training becomes more stable.

To this end, combining Eq. (2) and Eq. (4) results in our modulated transformation module (`MTM`) with the negligible computational overhead. Importantly, no additional explicit supervisions are required for the learning of `MTM`, making it as a plug-and-play module for various generators and tasks.

## 3.2 Content generation framework

Considering that generative adversarial networks (GANs) have advanced 2D/3D-aware image and video generation, we incorporate the proposed module into multiple popular generators for various synthesis tasks. Briefly, GANs formulate content generation as a two-player game where a generator $G(\cdot)$ learns the mapping from a pre-defined distribution to the data distribution while a discriminator $D(\cdot)$ aims at distinguishing the synthesized from the real data distribution. As no modifications are made on the discriminator, we thus neglect the relevant discription in the following context for brevity.

**Synthesizing 2D images.** A style-based generator is proposed in [23], which injecting the mapped latent codes into the generator layer by layer. Such layer-wise design of the architecture results in the semantic hierarchy [23, 56] where various layers usually encoder different visual concepts, making it become a standard generator for image synthesis. We thus choose the StyleGAN2 [25] and its alias-free version StyleGAN3 [26] as the image generator. To incorporate the proposed `MTM`, we simply replace one convolution at various resolutions with our module. In this way, the improved generators are able to handle the geometric variation from small to the large scales.

**Synthesizing 3D-aware images.** With the development of 2D image synthesis, researchers pour their attention to incorporate the inductive bias into the original generator such that more explicit control could be empowered. For instance, EG3D [7] makes the best of 2D image generators to produce three-views drawing of a 3D representation space, leading to a effective and efficient tri-plane representations for 3D-aware image synthesis. Accordingly, we could still insert the proposed `MTM` into the 2D generator of EG3D [7] (as before), indirectly rearranging the tri-plane representations.

**Synthesizing videos.** When producing videos, large geometric variation does not only exist across video instances but also occur over time. `MTM` is thus employed into the video generator. In particular, `MTM` is equpped with the previous state-of-the-art video generation approach StyleSV [61] that manages to lift a 2D generator to synthesize infinite and smooth video frames. Particularly, regular convolutional layers would be also replaced by our module. Furthermore, following the similar philosophy of [61], the latent code that drives the learnable offsets in Eq. (4) consists of the motion and content information simultaneously for video generation.

## 3.3 Training

**Learning objectives.** As is mentioned before, `MTM` makes no additional modifications on the discriminators and requires no explicit supervisions. Therefore, learning objectives strictly follow the corresponding approaches. For example, non-saturating logistic loss [13] together with $R_1$ regularization [30] is adopted for the adversarial training of the generator and discriminator.

Table 1: **Comparisons on 2D image generation.** FID, CLIP-FD and sFID are reported for the quantitative measurement.

(a) Evaluation on ImageNet-128 [10].

| Methods | FID↓ | CLIP-FD↓ | sFID↓ |
|---|---|---|---|
| StyleGAN2 [24] | 21.14 | 35.60 | 4.52 |
| *w/* MTM | **19.16** | **33.92** | **4.39** |

(b) Evaluation on TaiChi-256 [41].

| Methods | FID↓ | CLIP-FD↓ | sFID↓ |
|---|---|---|---|
| StyleGAN3 [26] | 21.36 | 25.97 | 4.79 |
| *w/* MTM | **13.60** | **19.27** | **3.57** |

(c) Evaluation on LSUN Church-256 [58].

| Methods | FID↓ | CLIP-FD↓ | sFID↓ |
|---|---|---|---|
| StyleGAN2 [24] | 4.04 | 33.34 | 8.15 |
| *w/* MTM | **2.32** | **25.98** | **6.93** |

(d) Evaluation on LSUN Cat-256 [58].

| Methods | FID↓ | CLIP-FD↓ | sFID↓ |
|---|---|---|---|
| StyleGAN2 [24] | 6.87 | 24.31 | 6.06 |
| *w/* MTM | **5.92** | **19.89** | **5.70** |

**Parameter efficiency.** Compared to the regular convolution, the newly-introduced parameters by MTM are mainly those of the extra convolution that aims at predicting the offsets. Importantly, as mentioned in Sec. 3.1, the number of output dimension is quite limited (18 in practice). Therefore, negligible parameters would be caused even counting extra parameters of multiple MTM in total.

**Computing efficiency.** Although all convolutions could be replaced by the proposed MTM (still causing negligible extra parameters), we observe that the training would be further slowed down due to a large number of new operations. Intuitively, the high resolution intermediate representations might have already determined the geometric structure. Inspired by this, we further investigate which layers are supposed to be enhanced by the proposed MTM. Our experiments suggest that replacing regular convolutions at few low-resolution layers is sufficient, leading to a comparable training efficiency. This also matches the finding in [56] that the low-resolution intermediate representations mainly control the spatial layout/structure.

## 4 Experiments

We evaluate the proposed MTM on various generative models for both image and video generation tasks. In Sec. 4.1, we outline the benchmarks, evaluation metrics, and corresponding baseline approaches involved in the experiment. Sec. 4.2 showcases the primary synthesis outcomes for 2D and 3D-aware image generation. We further integrate MTM into video generators in Sec. 4.3. Finally, we present the ablation study in Sec. 4.4 to demonstrate the impact of each design.

### 4.1 Settings

**Datasets.** We employ several challenging benchmarks to evaluate the efficacy of our module. Firstly, we use ImageNet [10], which contains approximately 1.2 million images covering 1000 object categories, showcasing significant geometric variation. Due to limited training resources, we resize all images to $128 \times 128$ pixels. Additionally, we utilize LSUN Church[58] and Cat [58], two typical benchmarks for image generation that feature outdoor scenes and non-rigid animals respectively. Furthermore, we employ TaiChi [41], which consists of around 3000 videos of people performing TaiChi, as an image benchmark. We treat each frame as an independent image for image generation since diverse gestures among individuals exist. All images are resized to $256 \times 256$ pixels.

Regarding video generation, we use SkyTimelapse [53], which captures moving clouds, and TaiChi [41], which features large non-rigid motion. Moreover, we use YouTube Driving[60], which collects 134 driving videos with varying weather conditions, regions, and cities. According to[61], these driving videos have relatively strict geometric constraints, making them an appropriate benchmark for our evaluation.

**Metrics.** We use Fréchet Inception Distance (FID) [17] as the quantitative metric for measuring the quality of image synthesis, as it can reflect human perception to some extent. We employ the official pre-trained Inception feature extractor, which can be downloaded from here. Additionally, we follow prior literature[28] and replace the Inception extractor with CLIP [35], which we refer to as CLIP-FD. Furthermore, to assess the synthesis quality from the perspective of spatial structure, we also report

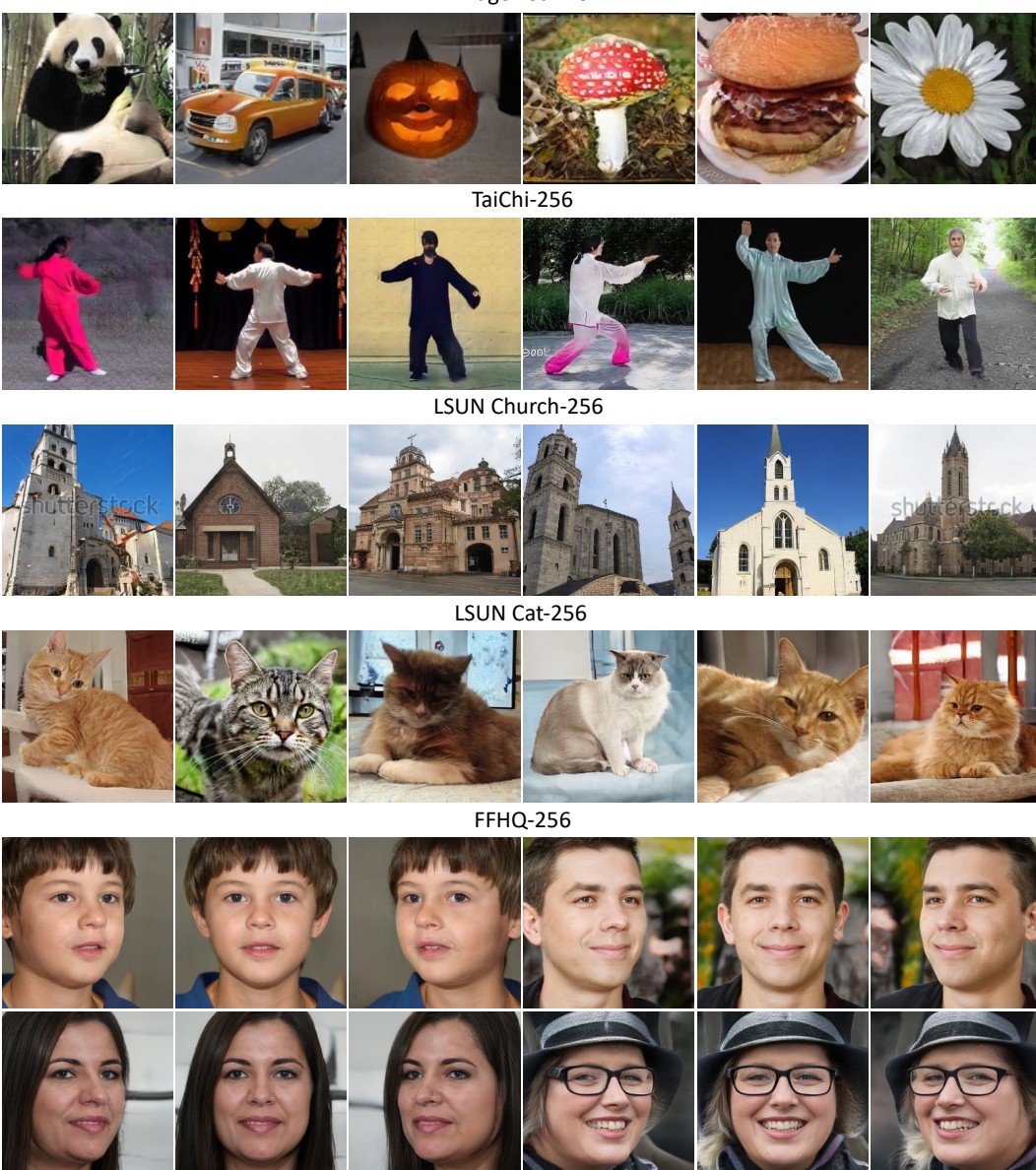

Figure 2: **Qualitative results** on various datasets for image generation. Note that synthesis on FFHQ-256 [23] is produced by a 3D-aware generator *i.e.*, EG3D [7].

sFID [31]. For video generation, we utilize Frechet Video Distance (FVD) [46] as the metric, with two temporal spans: consecutive 16 and 128 frames, respectively.

**Baselines.** For 2D and 3D-aware image generation, we select StyleGAN2 [24], StyleGAN3 [26], and EG3D [7] as the baseline approaches. We also employ StyleSV [61], the recent state-of-the-art video generator, as a baseline for video synthesis. Importantly, when incorporating our MTM with these baselines, we make no additional modifications to them. The training scheme and hyper-parameters remain untouched.

## 4.2 Towards image generation

**Main results.** Tab. 1 presents the experimental results on multiple datasets for 2D image generation. Notably, we treat all frames in TaiChi [41] as independent images. We keep all training hyper-

Table 3: **Comparisons on video generation.** $FID_\downarrow$ and $FVD_\downarrow$ are reported for the quantitative measurement.

(a) Evaluation on SkyTimelapse [53].

| Methods | $FVD_{16}$ | $FVD_{128}$ | FID |
|---|---|---|---|
| MoCoGAN [45] | 85.9 | 272.8 | - |
| DIGAN [59] | 83.1 | 196.7 | - |
| LongVideoGAN [5] | 116.5 | 152.7 | - |
| StyleGAN-V [42] | 73.9 | 248.3 | **40.8** |
| StyleSV [61] | 49.0 | 135.9 | 49.9 |
| *w/* MTM | **42.3** | **124.6** | 44.5 |

(b) Evaluation on TaiChi-256 [41].

| Methods | $FVD_{16}$ | $FVD_{128}$ | FID |
|---|---|---|---|
| MoCoGAN-HD [44] | 144.7 | - | - |
| DIGAN [59] | 128.1 | - | - |
| TATS-base [12] | 94.6 | - | - |
| StyleGAN-V [42] | 152.0 | 267.3 | 33.8 |
| StyleSV [61] | 97.4 | 188.9 | 26.6 |
| *w/* MTM | **89.5** | **180.6** | **25.2** |

(c) Evaluation on YouTube Driving [60].

| Methods | $FVD_{16}$ | $FVD_{128}$ | FID |
|---|---|---|---|
| StyleGAN-V [42] | 449.8 | 460.6 | 28.3 |
| StyleSV [61] | 207.2 | 221.5 | 19.2 |
| *w/* MTM | **194.8** | **198.4** | **10.3** |

parameters (*e.g.*, batch size, learning rates, number of iterations, and coefficients of $R_1$ penalty) identical for both baseline generators and the improved ones (*i.e.*, *w/* MTM).

Our module introduces consistent gains from all measurement perspectives. The differences in FID and CLIP-FD metrics suggest a significant improvement in overall synthesis quality. Specifically, without increasing the generators' capacity, FID on ImageNet can be boosted from 21.14 to 19.16. Previous approaches [38, 3] suggest that capacity matters for large-scale datasets. However, our negligible computations also result in a noticeable synthesis improvement, strongly demonstrating our module's effectiveness to some extent. Moreover, sFID indicates that generators can produce images with more reasonable spatial structure when equipped with our MTM. Fig. A1 showcases the qualitative results for 2D and 3D-aware image generation.

Tab. 2 presents the experimental results for 3D-aware image generation. It is important to note that we strictly follow the data pre-processing of the original EG3D [7]. Although EG3D runs on well-aligned facial datasets, such as FFHQ [23], which hardly contain large-scale geometric variation, we still observed an improvement in quality. This further narrows the gap between pure 2D image generation (*i.e.*, StyleGAN2 [24]) and 3D-aware image generation (*i.e.*, EG3D [7]).

Table 2: **3D-aware image generation.**

| FFHQ-256 | $FID_\downarrow$ |
|---|---|
| StyleGAN2 [24] | 3.78 |
| EG3D [7] | 4.32 |
| *w/* MTM | **4.07** |

## 4.3 Towards video generation

**Main results.** Tab. 3 presents the comparisons for video generation. Additionally, Fig. A2 showcases the qualitative results. Notably, StyleSV [61] serves as our baseline. Specifically, StyleSV trains a video generator in two stages, first pre-training an image generator on video datasets where each frame is considered an individual image. As expected, generators with our module outperform the baselines again. After pre-training, StyleSV finetunes the generator on video datasets, incorporating temporal motion representations and explicit temporal modeling on the generator and discriminator, respectively. Our MTM is always trainable during these two stages. Finally, we compare the improved StyleSV [61] against prior literature on multiple benchmarks.

Our MTM significantly improves the generator's performance on video synthesis benchmarks, leading to new state-of-the-art results. The improvement is evident from both FVD16 and FVD128 metrics, indicating enhanced short-term and long-term video synthesis. Additionally, our module improves the single-frame quality, as evidenced by lower FID scores. In summary, our proposed MTM enables the original generator to handle large geometric variation across instances and over time, resulting in significant improvements in video synthesis performance.

SkyTimelapse

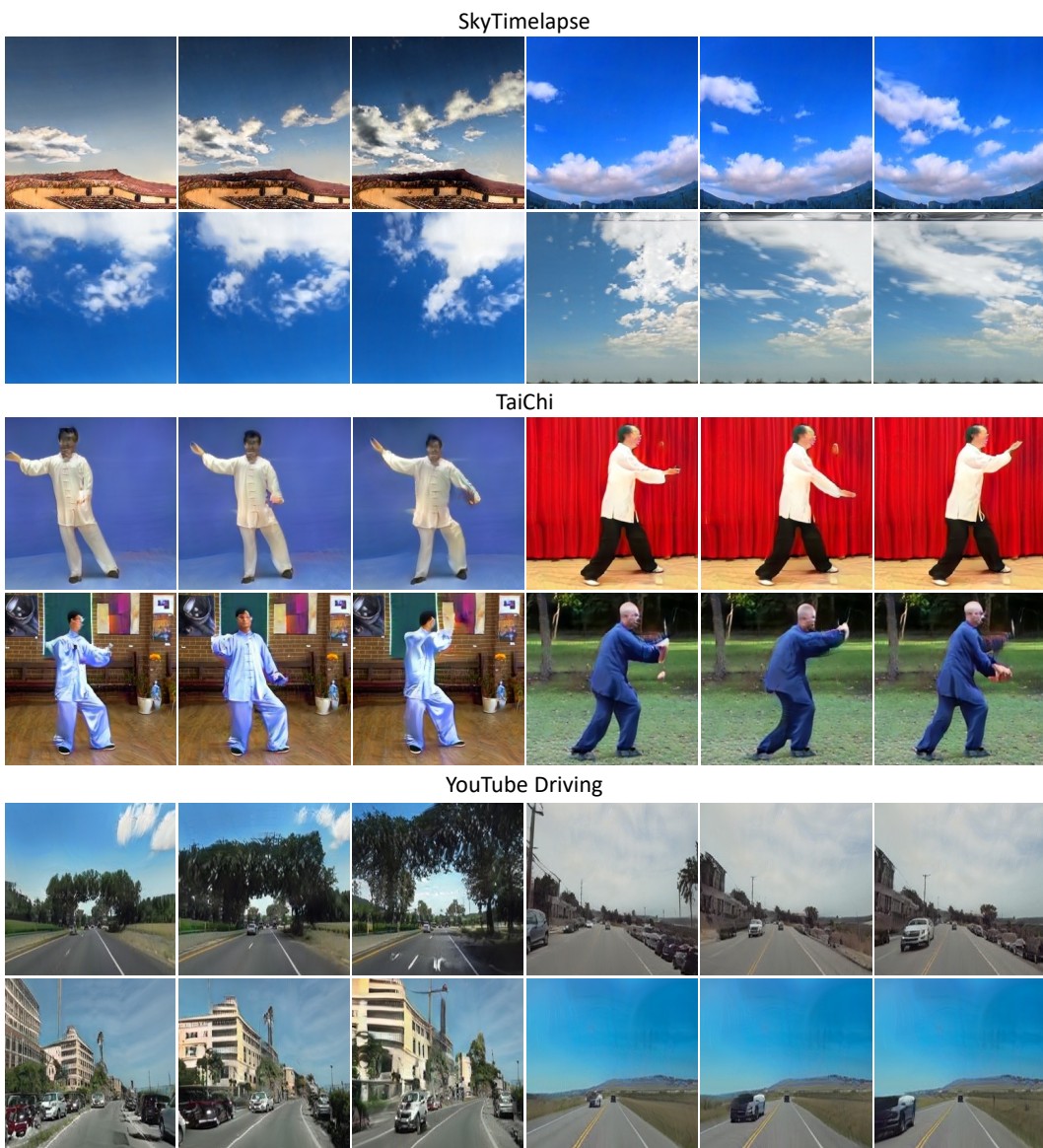

TaiChi

YouTube Driving

Figure 3: **Qualitative results** on various datasets for video generation. Due to the limited space, we subsample frames from the generated videos here.

## 4.4 Ablation studies

**Which layer requires the proposed** `MTM`**?** It is possible to replace all convolutions in a given generator with our module, but doing so would significantly slow down the training process. Therefore, we conducted an ablation study to investigate the best trade-off between speed and performance. Prior literature [23, 56] suggests that a synthesized image is represented hierarchically. Specifically, the lower-resolution layers (closer to the latent codes) tend to determine the spatial structure of a generated image, while the higher-resolution layers focus on fine-grained details like texture. This inspired us to question whether replacing all convolutions is necessary. Instead, we hypothesized that empowering the layers most relevant to the spatial structure might be sufficient. To test this hypothesis, we split the layers of the generator into three groups based on the resolution of intermediate representations: low, mid, and high.

Tab. 4 presents the comparisons on ImageNet [10] when replacing convolutions of different layers with our `MTM`. The results indicate that applying our module at the low-resolution layers achieved appealing performance with acceptable training time. Incorporating more modules into the generator

Table 4: **Ablation studies** on effect of various layers. All layers of the generator are split into three groups: Low, Mid and High which we replace with the proposed module. Obviously, introducing it into the low-resolution layers results in the best trade-off between performances and training efficiency.

| Low | Mid | High | FID$_\downarrow$ | CLIP-FD$_\downarrow$ | sFID$_\downarrow$ | Training time | Inference time | # Param. (MB) |
|:---:|:---:|:---:|:---:|:---:|:---:|:---:|:---:|:---:|
| | | | 21.14 | 35.60 | 4.52 | 1.0× | 1.0× | 27.78 |
| ✓ | | | **19.16** | **33.92** | **4.39** | 1.2× | 1.0× | 28.55 |
| ✓ | ✓ | | 19.35 | 34.53 | 4.42 | 2.1× | 1.0× | 28.94 |
| ✓ | ✓ | ✓ | 20.02 | 34.76 | 4.46 | 3.2× | 1.0× | 29.32 |

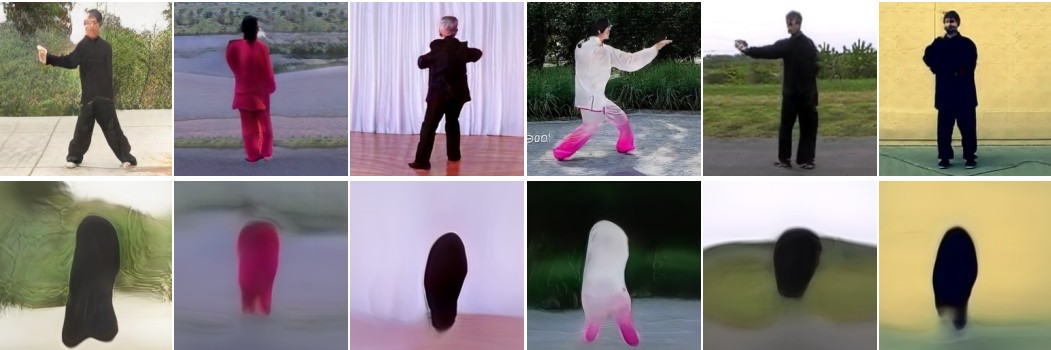

Figure 4: **Visualization** of disabling offsets after training. We set the learnable offsets as zero at the second row. Interestingly, the shape of the main objects in the synthesis becomes quite similar, demonstrating the potential of MTM in performing *explicit* deformation to deliver various shapes from the blob-like.

did not lead to further improvements but significantly slowed down the training process. We also observed that training became somewhat unstable when using multiple modules on high-resolution layers in practice. Therefore, we decided to use MTM only in the low-resolution layers, which is also how we implemented all image and video generation experiments presented above.

**What do the learnable offsets exactly capture?** Visualizing the learnable offsets of our MTM is challenging and non-intuitive because it operates directly on intermediate representations with the generator. Therefore, we disabled the learnable offsets to analyze the behavior of MTM. Specifically, after training, we set the offsets to zero. Importantly, even when the learnable offsets are set to zero, the regular convolutions still function normally.

Fig. 4 presents the corresponding results. The first row shows the original synthesis for image generation on TaiChi [41]. When we roughly disabled the learnable offsets, the synthesis seemed to suffer from collapse. Interestingly, the shape of the main objects in the synthesis became blob-like. This demonstrates that our proposed MTM performs explicit deformations/transformations to deliver a variety of gestures that all derive from a blob-like shape.

## 5 Conclusion

In this work, we investigate how GANs handle data collection that involves multiple samples with large-scale geometric variation. Since regular convolution is inherently limited, we propose a new method called the Modulated Transformation Module (MTM) that enables explicit and flexible transformations of intermediate representations. Through careful design and comprehensive study, our method significantly improves the quality of image and video generation without additional supervision or computational overhead. We evaluate our method across multiple benchmarks and architectures, demonstrating its effectiveness in various datasets and generative tasks. We believe that MTM could become a fundamental operation in future generative foundation models.

**Discussion.** While our proposed MTM consistently leads to gains for both image and video generation, there are still some limitations that we would like to discuss. First, we have not explored whether our method could result in further improvements for other generative models, such as auto-regressive models and score-based diffusion models. Fortunately, recent work [39, 21] has already demonstrated

that GANs are comparable to other generative models with the proper scaling scheme. Second, the effectiveness of our `MTM` on large-scale generation tasks such as text-to-image generation, which often requires a larger GAN, is unknown. Given the current limited resources, we leave this study for future work.

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

## Appendix

This supplementary material is organized as follows. We first discuss the broader impact of the proposed method in appendix A. Besides, appendix B provides the implementation details including required resources and codebases we used. Fig. A1 and Fig. A2 shows more synthesis. At last, appendix C presents more comparisons and experimental results.

## A    Broader impact

The proposed approach represents a significant advancement in the field of image and video synthesis, which is both a fundamental and practical problem. The introduction of the modulated transformation module has greatly enhanced the capacity for handling geometric variation, making generative models more accessible to the general public. However, this technique may also facilitate the creation of fake content, such as Deepfake, which could have negative consequences. We want to emphasize that we strongly oppose the misuse of this approach to violate security and privacy issues. The negative impact of such misuse can be mitigated by the development of deep fake detection technology.

## B    Implementation details

MTM is developed based on the official implementation of StyleGAN3 and StyleGAN-V for 2D image and video generation. Besides, regarding the 3D-aware image generation, we use the official implementation of EG3D. That is, we strictly follow the data-preprocessing and hyper-parameters of the original ones. Although we train each model on a server with 8 A100 GPUs, it requires no large GPU memory and allows other researchers to reproduce easily.

## C    More Results

**Quantitative results for image generation.** Tab. A1 provides more results for the image synthesis. In particular, frame in the video datasets used for video generation is regarded as an individual image. FID is reported for the comparison. Obviously, our method could still improve the baseline generator.

**Longer training for the baseline.** To investigate the performances introduced by our approach, we train the baseline methods for double training iterations. Tab. A2 presents the comparison where our method remains to outperform baselines, even with longer training. Interestingly, the synthesis of the baseline on ImageNet becomes worse with longer training while our module could significantly improve the FID with the same training iterations and negligible computational overhead, demonstrating the effectiveness of our method.

Table A1:  **Quantitative results on StyleGAN3.** FID (the lower the better) is reported for the comparison.

| Methods | SkyTimelapse | TaiChi | YouTube Driving |
|---------|--------------|--------|-----------------|
| StyleGAN3 | 9.37 | 21.36 | 8.40 |
| *w/* MTM | **8.72** | **13.60** | **6.16** |

Table A2:  **Comparison against longer-tuned baseline.** FID (the lower the better) is reported for the comparison.

| Methods | Iterations | ImageNet | LSUN Church | LSUN Cat |
|---------|-----------|----------|-------------|----------|
| StyleGAN2 | 1× | 21.14 | 4.04 | 6.87 |
| StyleGAN2 | 2× | 22.35 | 2.82 | 6.38 |
| MTM | 1× | **19.16** | **2.32** | **5.92** |

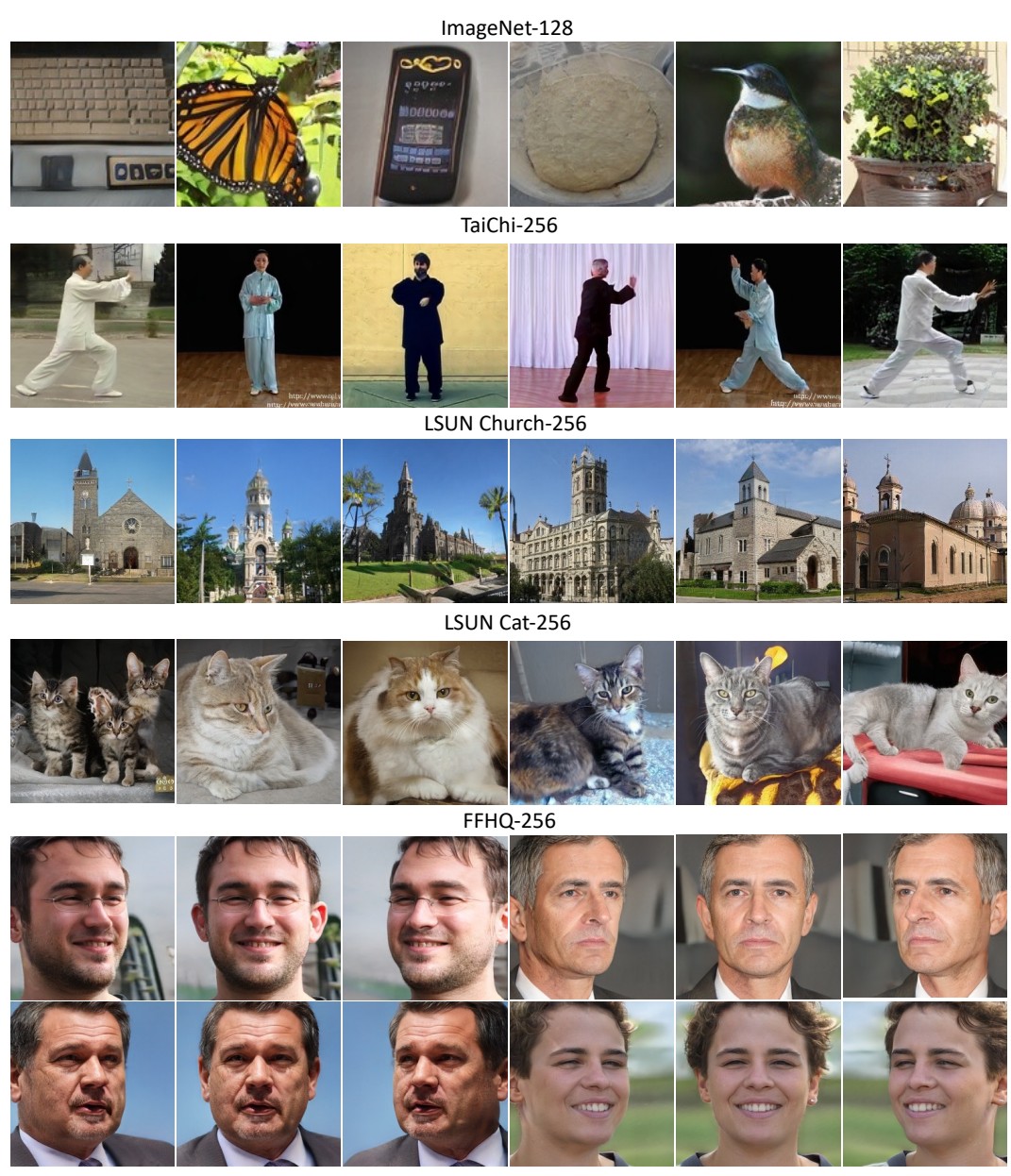

Figure A1: **Qualitative results** on various datasets for image generation.

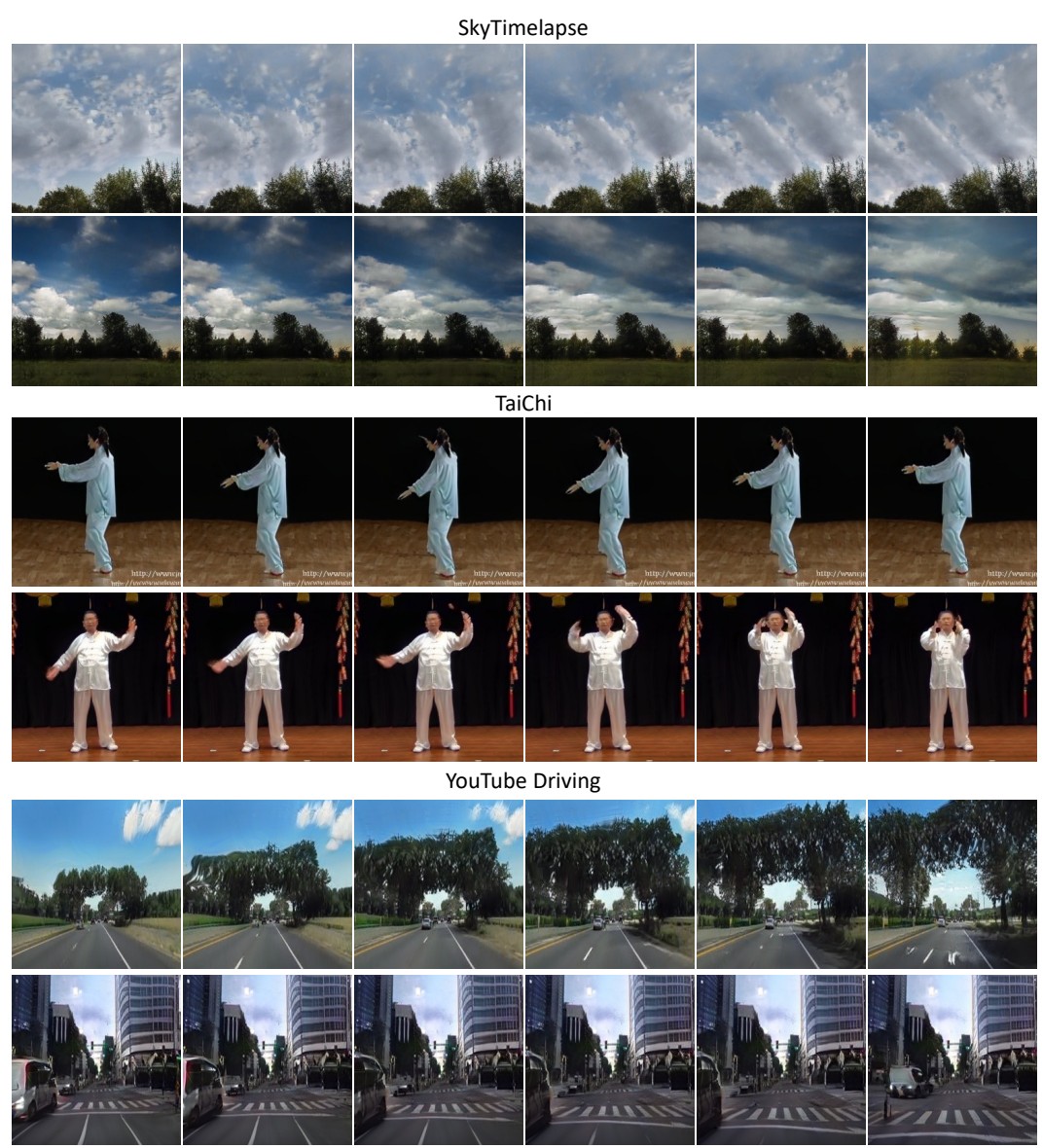

Figure A2: **Qualitative results** on various datasets for video generation.

