# OpenReview forum: "Learning Modulated Transformation in GANs"
_NeurIPS.cc/2023/Conference — NeurIPS 2023 poster_

### Official Review · Reviewer_A6mm · 2023-07-02

**Soundness:** 3 good
**Presentation:** 3 good
**Contribution:** 3 good
**Rating:** 7
**Confidence:** 4

**Summary:**

The paper applies ideas from Spatial Transformer Networks in the context of generative models, specifically GANs, introduces a new module that gives the GAN generator a more natural way to generate content at spatially varying locations. The resulting models are better able to generate content with smooth geometric changes, such as videos, but more traditional non-video datasets also see an improvement. The method seems easy to implement and shows promise when applied to various network architectures and datasets.

**Strengths:**

The changes introduced are self-contained and minimally disruptive, and notably do not require changes to hyper parameters or training schemes. The numerical results look good across the board, and the technique seems compatible with various network architectures.

Even though the proposed method seems straight-forward to implement, the promised code release is still a plus.

**Weaknesses:**

The paper does not impart much intuition to the reader: what exactly are the new capabilities of these models? Are there smooth spatial movements upon latent-space interpolation etc? Are there any situations where the proposed method performs worse and should not be used? In a similar vein, visual comparisons to baselines are missing. Seeing FIDs decrease is nice, but understanding the properties of the new model is even better. Figure 4 is a step in this direction, but more is needed.

Figure 2 is quite uninformative on its own - without comparisons to baseline models, the reader doesn't really learn anything (we already know vanilla SG2 works great for cats, for example).

Failure cases are not discussed in detail - were there datasets where the proposed method did not help, or even hurt performance?

**Questions:**

Line 42 talks about warping the input features, whereas line 112 talks about bilinear interpolation (presumably of the unwrapped input feature map). Figure 1 also seems to indicate that the feature map remains unwarped. How is the operation implemented in practice? The text and figures need to be unambiguous.

SG3 takes great care to use very high-quality resampling operations within the network, in order to minimize aliasing. Does the proposed bilinear interpolation negatively affect the equivariance of SG3? Is the interpolation operation otherwise compatible with the strict signal-processing requirements of SG3? A comment on this would be valuable.

**Limitations:**

The limitations are not discussed much and should be expanded upon. The paper makes it seem like the proposed method lead to improvements in every single setting that was tried, which is unlikely.

---

> ### Author Rebuttal · Authors · 2023-08-09
>
> **Q1. "What exactly are the new capabilities of these models?"**
>
> Compared to existing GANs that perform convolution at *fixed* locations (*i.e.*, shared by all samples), we offer the generator an addition degree of freedom though performing convolution at *variable* locations. For this purpose, we apply learned transformation to feature maps before feeding them to the convolution operation, where the learned transformation is controlled by the latent code. Such a newly introduced lightweight plug-in alleviates the difficulty of modeling geometric variations within the dataset. Here, we would like to clarify that our design does not intend to enable new functionalities, but it is indeed possible to study whether MTM enables some functional byproducts during inference. Thanks for your suggestion and we will leave it as future work.
>
> **Q2. "Figure 2 is quite uninformative on its own - without comparisons to baseline models."**
>
> Thanks. We include some qualitative comparisons in Fig. R2 (see **the newly uploaded one-page PDF**). We will also add the results in revision.
>
> **Q3. "Failure cases are not discussed in detail - were there datasets where the proposed method did not help, or even hurt performance?"**
>
> Thanks. So far, we evaluate our approach on multiple datasets for image generation, video generation, and 3D-aware image synthesis, and observe consistent performance gain. Still, it is interesting to mention that, on the commonly used FFHQ dataset, applying our MTM seems to have neither positive nor negative effect. One possible guess is that FFHQ contains human faces that are with similar shape and already well-aligned, containing limited geometric variations. Under such a case, using MTM would bring additional computation overheads during training yet achieve on-par performance. We will add the discussion.
>
> | FFHQ-256 | FID |
> | :- | :-: |
> | StyleGAN2 | 3.72 |
> | *w/* MTM  | 3.74 |
>
> **Q4. "How is the operation implemented in practice?"**
>
> In practice, feature warping is incorporated into the convolutional operation. Namely, when a kernel performs convolution, the desired features are obtained through bilinear interpolation, which is equivariant to warping them first. We will revise the text and the figures to clarify this. Also, the core implementation (PyTorch code) is already submitted as the supplementary material, and we will release the entire code to facilitate reproduction.
>
> **Q5. "Does the proposed bilinear interpolation negatively affect the equivariance of SG3?"**
>
> Thanks. Following the suggestion, we evaluate the generator with respect to its equivariance property to translation (measured by EQ-T, where higher number is better). The table below suggests that our MTM significantly improves the synthesis performance without sacrificing much equivariance property. We will add the results.
>
> | TaiChi-256 | FID | EQ-T |
> | :- | :-: | :-: |
> | StyleGAN2 + Fourier features | 23.89 |  9.36 |
> | StyleGAN3                    | 21.36 | 45.87 |
> | *w/* MTM                     | 13.60 | 41.52 |

---

> > ### Comment · Reviewer_A6mm · 2023-08-13
> >
> > Thank you for the thorough response.
> >
> > As for Q1, I'm not confused about what the new capabilities are from a technical standpoint. What I'm wishing for is some intuition about the properties of these new models. As the authors of a new architecture, I expect you to play around with the models and try to figure out if they behave differently compared to the baseline. Maybe latent space interpolations show smoother movements of objects etc. As it stands, the paper seems to indicate that the FIDs improve, but that the models otherwise behave identically to the baselines.
> >
> > The other questions have been addressed, thanks.
> > I will follow the discussions here and reconsider my rating.

---

> > > ### Author Response · Authors · 2023-08-13
> > > **Discussion**
> > >
> > > Thanks for your reply, together with the clarification on Q1. Your suggestion about "playing around the model supported by our new architecture to study whether it behaves differently compared to the baseline" is very instructive. Following your suggestion, we evaluate latent space interpolation with the model well-trained with our MTM, and report the comparison results against the baseline in the table below. We can tell that our method also helps improve the interpolation performance. We will also add some qualitative results in the revision (not sure whether this is allowed in the discussion period).
> > >
> > > | TaiChi-256 | FID | FID after Interpolation |
> > > | :- | :-: | :-: |
> > > | StyleGAN3 | 23.89 | 22.64 |
> > > | *w/* MTM  | 13.60 | 14.51 |
> > >
> > > However, we need to admit that, taking the improved synthesis performance (*i.e.*, from 23.89 to 13.60) into account, we cannot directly conclude whether the "smoother movements" come from a more capable generator or from a better interpolation property. More detailed analyses would be needed and we leave them as the follow-up work. You are right that our work, in its current form, primarily focuses on how to improves GANs in learning from data with large geometry variations. We believe that *this problem itself is already challenging and fundamental in GAN studies, to which we provide an effective and generalizable solution*.

---

### Official Review · Reviewer_zzjJ · 2023-07-05

**Soundness:** 3 good
**Presentation:** 3 good
**Contribution:** 2 fair
**Rating:** 4
**Confidence:** 5

**Summary:**

The paper introduces learnable modulated convolutions to the literature on GANs. Instead of using standard convolutions with fixed 3x3 kernels, the paper proposes to learn the kernel spatial offsets to allow flexibility in the receptive field of generators. The proposed module can be added to any GAN based on convolutions, and experiments demonstrate it for StyleGANv2, StyleGANv3, EG3D, StyleSV. The module introduces only marginal computational overhead but improves the synthesis of models substantially.

**Strengths:**

- To my knowledge, this paper brings deformable convolutions to the GAN literature for the first time. This method can be easily applied to almost any CNN-based GAN. It introduces only marginal computational overhead but sometimes gives big gains if FID. It can play a role in the further development of GANs, esecially in video synthesis.
- The paper is easy-to-follow, the ideas are intuitive, and the experiments are extensive.

**Weaknesses:**

[Novelty] - I do not understand what is the technical novelty introduced in Sec. 3. Learnable offsets, as shown in Sec. 3.1., have already been introduced in cited prior work (e.g., DCN). The Style Block used for latent modulation in Sec 3.2. has already been introduced in Karras et al. It seems that the method naively applies existing technology (DCN) into existing GANs without any modifications or novel analysis.  It is therefore not clear where is the technical challenge the paper solves.

[Comparison] - The presented motivation for introducing learnable offsets is the fact that GANs are too "local" and cannot handle more global dependencies. This problem has been also investigated in prior work (U-Net based discriminator, SAGAN, etc), and some solutions exist. With this motivation, I would expect a discussion and comparison of how well global coherency is preserved in generated images thanks to the introduced method, as well as comparisons to alternatives.

[Comparison] - I would also expect more visualizations of what the module actually learns. For example, how big are the learned offsets usually in comparison to whole feature dimensions?

[Comparison] - As a thought, would a simpler strategy without learnable parameters that increases the receptive field of convolutions (e.g., 5x5, 7x7 convolutions, strides) also help?

[Experiment] - At the moment, there are no visual results to see the qualitative effect of the presented method on studied models (e.g., StyleGANv2 w/o ours vs with ours). I think this is an interesting yet missing analysis.



**Questions:**

Please comment on the Weaknesses.

In addition, why are the baselines in Table 1 non-uniform? Would it be possible to show results for all the datasets for both StyleGANv2 and StyleGANv3?

**Limitations:**

Limitations and Societal Impacts are addressed.

---

> ### Author Rebuttal · Authors · 2023-08-09
>
> **Q1. About the technical novelty introduced in Sec.3, and "It seems that the method naively applies DCN into existing GANs without any modifications of novel analysis. It is therefore not clear where the technical challenge the paper solves".**
>
> Disagree. Unlike DCN, which is originally evaluated on discriminative tasks, MTM is particularly designed for generative tasks, which require to decode the sampling stochasticity to diverse realistic images. To this end, we condition the learning of the transformation on the sampled latent code. As the table shown below, directly applying deformable convolution to GANs drastically harms the synthesis performance. By contrast, applying MTM to GANs brings substantial performance gain. From this perspective, even though DCN and MTM share a similar philosophy, MTM surpasses DCN and works as an general and efficient module in GANs, which is further supported by the experiments on a range of generation tasks (including image generation, video generation, and 3D-aware image synthesis) and datasets. To our best knowledge, we are the first to *make the idea of spatial transformation work on generative models*, which is not that straightforward according to the table below. We hope our discovery could make MTM a basic operation in the future design of GANs. We will add the additional results and the discussion in the revision.
>
> | TaiChi-256 | FID |
> | :- | :-: |
> | StyleGAN3                  |  21.36 |
> | *w/* deformale convolution | 192.65 |
> | *w/* MTM                   |  13.60 |
>
> **Q2. "I would expect a discussion and comparison of how well global coherency is preserved in generated images thanks to the introduced method, as well as comparisons to alternatives."**
>
> In generation tasks, global coherency should be well reflected by the overall synthesis quality, which is usually measured by FID and FVD. The consistent FID/FVD improvement on various tasks and datasets provides strong support for the effectiveness of our proposed MTM. Following your suggestion, we compare our approach with SAGAN, which introduces self-attention to the GAN generator. For a fair comparison, we use StyleGAN3 as the same backbone and enhance the conventional convolution operation with self-attention or our MTM. The table below demonstrates the superiority of MTM over SAGAN.
>
> | TaiChi-256 | FID |
> | :- | :-: |
> | StyleGAN3  | 21.36 |
> | *w/* SAGAN | 26.83 |
> | *w/* MTM   | 13.60 |
>
> **Q3. "I would also expect more visualizations of what the module actually learns. For example, how big are the learned offsets usually in comparison to whole feature dimensions?."**
>
> Due to the lack of spatial correspondence between feature maps and the final synthesis, it is hard to directly visualize the offsets learned by our MTM, which are at early layers (*i.e.*, small resolution), on the generated image. We have included Fig. 4 in the manuscript, where the offsets are disabled during inference, to give a rough overview of what the module learns. Following your suggestion, we summarize the stats of the learned offsets at the resolution of 36x36 and 52x52. As shown below, compared to the conventional convolution which adopts a fixed receptive field, our MTM offers the model an additional degree of freedom to decode the sampling stochasticity.
>
> | TaiChi-256 | 36x36 | 52x52 |
> | :- | :-: | :-: |
> | StyleGAN3 |   3 ± 0    |   3 ± 0    |
> | *w/* MTM  | 4.7 ± 0.22 | 5.4 ± 0.36 |
>
> **Q4. "As a thought, would a simpler strategy without learnable parameters that increases the receptive field of convolutions (e.g., 5x5, 7x7 convolutions, strides) also help?"**
>
> Following your suggestion, we conduct an experiment by replacing the 3x3 convolutional kernels in StyleGAN3 with 5x5 kernels. The results are listed below, where we can tell that larger kernels do not always lead to better performance.
>
> | TaiChi-256 | FID |
> | :- | :-: |
> | StyleGAN3            | 21.36 |
> | *w/* 5x5 convolution | 45.94 |
> | *w/* MTM             | 13.60 |
>
> **Q5. "There are no visual results to see the qualitative effect of the presented method on studied models."**
>
> Thanks. We include some qualitative comparisons in Fig. R2 (see **the newly uploaded one-page PDF**). We will also add the results in revision.
>
> **Q6. "Why are the baselines in Table 1 non-uniform?"**
>
> Compared to StyleGAN2, StyleGAN3 targets the anti-alias property of the generator instead of improving the synthesis performance. In fact, StyleGAN2 has already achieved satisfying performances on many object-centric datasets, such as LSUN church and LSUN cat. But for the challenging TaiChi dataset, where the human body is far from aligned, StyleGAN2 struggles in learning such a complex distribution. Hence, we choose StyleGAN3 as the baseline on TaiChi dataset considering its effectiveness in learning from unaligned data. We also evaluate the generator with respect to its equivariance property to translation (measured by EQ-T, where higher number is better). The table below suggests that our MTM significantly improves the synthesis performance without sacrificing much equivariance property. Due to the limited time, however, we cannot conduct experiments on all datasets with both StyleGAN2 and StyleGAN3.
>
> | TaiChi-256 | FID | EQ-T |
> | :- | :-: | :-: |
> | StyleGAN2 + Fourier features | 23.89 |  9.36 |
> | StyleGAN3                    | 21.36 | 45.87 |
> | *w/* MTM                     | 13.60 | 41.52 |

---

> > ### Comment · Reviewer_A6mm · 2023-08-13
> >
> > I share some of reviewer zzjJ's concerns about novelty (Q1): the paper proposes delta_p = ModConv(x, z) (eq. 4) as the modulation mechanism, but the ModConv operation itself is not new, only the context in which it is applied. Was any testing performed to make sure the proposed mechanism is indeed better performing than alternatives (that also incorporate the latent vector)? As it stands it does indeed seem like two existing techniques were combined without much modification or analysis.

---

> > > ### Author Response · Authors · 2023-08-13
> > > **Discussion on Novelty**
> > >
> > > Thanks for your comments. We hope the following discussions could help address the novelty concern.
> > >
> > > - Our MTM is **ideationally novel** in that we introduce a lightweight plug-in module into GANs and bring consistent performance gain across a range of architectures and datasets. The nice experimental results are appreciated by all reviewers, demonstrating the effectiveness and generalizability of our MTM. Hence, we hope our MTM could **play a fundamental role** in the future design of GANs, just like AdaIN raised in StyleGAN. We believe our discovery, which is **never explored before**, would be of great interest to most audiences in the field of GANs.
> > >
> > > - Our MTM is **logically well motivated** from the aspect of geometry variation modeling. To offer the generator in GANs an additional degree of freedom to handle the geometry variation, we propose to introduce instance-aware learnable offsets to the convolution operation. In practice, this idea shares a similar philosophy to DCN (*i.e.*, ModConv()), which has already been well evaluated and efficiently implemented in many discriminative tasks. Therefore, we choose to borrow the implementation of DCN instead of designing a new approach for offset learning in convolution. We believe DCN should have already made many attempts and selected deformable convolution as its final form. Our contribution is to incorporate the stochasticity into the offset learning process, which is essential to solving generative tasks.
> > >
> > > - We would also like to argue that making deformable convolution compatible with GANs is **not a very straightforward thing** (see the table transcribed below). As appreciated by Reviewer zzjJ, our work is the first to demonstrate the effectiveness of deformable convolution in the GAN literature, which discovery we believe already serves as a strong contribution, let alone its simplicity and generalizability.
> > >
> > > | TaiChi-256 | FID |
> > > | :- | :-: |
> > > | StyleGAN3                  |  21.36 |
> > > | *w/* deformale convolution | 192.65 |
> > > | *w/* MTM                   |  13.60 |
> > >
> > > Finally, with all due respect, we would like to point out that most fundamental designs are simple (such as the residual connection in ResNet and AdaIN in StyleGAN) as long as they are well motivated and empirically work.

---

> > ### Comment · Reviewer_zzjJ · 2023-08-14
> > **Reply from Reviewer zzjJ on rebuttal.**
> >
> > I sincerely thank the authors for their answers.
> > I have no concerns about the performance gains that the proposed technology offers, and see potential for future usage in other MTM in other GANs.
> >
> > I believe, however, that a NeurIPS paper should not only demonstrate that some (inherited from non-GAN literature) technology brings big improvements in GANs, but also provide novel lessons and some insights for the community. My current evaluation of this aspect is not high.
> >
> >
> >
> > In this regard, my two major concerns are:
> >
> > 1.1) The proposed MTM is taken in DCN. The idea (Equations 1,2,3) is exactly the same as in DCN. From the implementation perspective, the paper incorporates DCN into StyleGAN's convolutions that were already part of StyleGAN convolutions.
> >
> > 1.2) I think that table Q1 is misleading. StyleGANs by default use noise modulation. In line 2 (w/deformable convolution), the noise modulation is deactivated. Therefore, the high FID of 192.65 just shows that having at least some form of noise modulation is important for StyleGANs. It is not surprising that DCN+modulation performs better than just DCN.
> >
> > 2.1) The paper lacks explanations about the observed results. I agree with reviewer A6mm that "intuition about the properties of these new models" is missing. I agree with the reviewer cDZK that the motivation for the module is explained vaguely and does not correspond to the results.
> >
> > 2.2) In the rebuttal, the answers for Q2 and Q4 are not informative. The results in the tables are not intuitive and not properly explained. Why using 5x5 convolutions is so harmful (while MTM also becomes somewhat 5x5 given the answer Q3)? I still cannot understand the intuition of why the FID is improved so greatly with MTM, while techniques with a similar motivation do not improve it at all.
> >
> > To conclude, I still do not understand the novelty and explanation of "why" it works.
> >
> > I would be happy to continue the discussion with the two following questions:
> > 1) Is it not true that the noise modulation being used comes straightforwardly from default StyleGAN modulation? If not, please explain the difference.
> > 2) Please provide a concise comprehensive explanation of *why* using DCM brings such big gains in FID.

---

> > > ### Author Response · Authors · 2023-08-14
> > > **Clarification on the Implementation**
> > >
> > > Thanks for your reply. After reading your further concerns, we think we have traced the gap between the message we want to deliver and your actual understanding. In the following, we would like to make some clarifications, which we will also include in the revision to avoid misunderstanding of readers.
> > >
> > > - First, we would like to reaffirm that there are **two** noise modulations in our proposed MTM, one for style modulation in the original StyleGAN and the other for transformation modulation, which distinguishes our MTM from the deformable convolution. Detailed explanations are as follows:
> > >
> > >     1. With $x$ and $z$ denoting the feature map and the latent code respectively, the conventional convolution can be formulated as $\texttt{Conv}(x)$.
> > >     1. Deformable convolution proposes to first learn an offset $t = \texttt{Conv}(x)$ from the feature map, which is then used to guide the convolution operation, written as $\texttt{DConv}(x, t) = \texttt{DConv}(x, \texttt{Conv}(x))$.
> > >     1. StyleGAN propose style modulation which uses the latent code to modulate the feature map, written as $\texttt{ModConv}(x, z)$.
> > >     1. In our MTM, we propose to incorporate deformation into GANs, where **both** the feature map and the learnable offsets are controlled by the latent code, as $\texttt{MTM}(x, z) = \texttt{DConv}(\texttt{ModConv}(x, z), \texttt{ModConv}(x, z))$. It is noteworthy that, the learnable offset in DCN, $t = \texttt{Conv}(x)$, differs from the **stochasticity-aware** learnable offset in our MTM, $t = \texttt{ModConv}(x, z)$.
> > >
> > > - Second, we would like to argue that the modification from learnable offset (*i.e.*, in DCN) to stochasticity-aware learnable offset (*i.e.*, in MTM) is fundamental and essential. Recall our motivation that the convolution kernel in $\texttt{Conv}$ and $\texttt{ModConv}$ only interacts with the feature map at **fixed** locations. We would like to offer the model an additional degree of freedom to handle the geometry variation. Hence, we do not only require the receptive field to be large, but more importantly we would like the receptive field to **vary across instances**. That is the reason why modulating the learnable offset with the latent code is important.
> > >
> > > - Third, you might have mis-interpreted the results in Q1. For the experiment "*w/* deformable convolution", we do *not* disable the original style modulation in StyleGAN. Instead, we compare $\texttt{DConv}(\texttt{ModConv}(x, z), \texttt{ModConv}(x, z))$ with $\texttt{DConv}(\texttt{ModConv}(x, z), \texttt{Conv}(x))$ to validate the effectiveness of changing $t = \texttt{Conv}(x)$ to $t = \texttt{ModConv}(x, z)$. That is the reason why we claim in the previous response that generative models may follow a different philosophy from discriminative models. Directly applying DCN to GANs causes strong training instability.
> > >
> > > - Fourth, for the explanation of results in Q2 and Q4, we believe that even self-attention and 5x5 kernel size could help enlarge the receptive field, but the field is still **fixed** among instances. Instead, MTM provides a solution for the generator to varying the receptive field across instances.
> > >
> > > Now, we will answer the reviewer's follow-up questions.
> > >
> > > 1. The noise modulation being used does not only come from the default StyleGAN modulation, but is also applied to offset learning.
> > >
> > > 1. Using DCM brings such big gains in FID is because we offer the generator an additional degree of freedom to handle the geometry variation with a varying receptive field across instances.
> > >
> > > 1. Main insights of this work:
> > >     - Instance-wise variation is important for the design of GANs, which is also a clear difference between generative models and discriminative models.
> > >     - The potential usage of MTM as a basic operation in the future design of GANs is already a sound contribution, which is also appreciated by the reviewer.
> > >
> > > We hope the above discussions could help address your concerns. Again, we will revise our manuscript according to our discussions to make the presentation clearer. Thanks again for your suggestions.

---

> > > > ### Comment · Reviewer_zzjJ · 2023-08-17
> > > > **Answer on clarifications**
> > > >
> > > > I thank the authors for their reply.
> > > >
> > > > ### Novelty
> > > >
> > > > - The authors correctly identified my misunderstanding. Based on the paper writing, I think it would be necessary to incorporate the three first items of the answer into the paper, since any other reader may also get confused in the same way.
> > > > - The authors highlighted the difference between naively applying DCN to GANs: the difference is that offsets are conditioned on both z and x. This increases my evaluation of novelty, although still not to a high degree. When a person skilled in GANs is asked to incorporate deformable convolutions into GANs, this would probably be the most obvious step after the straightforward implementation (with FIDof 192+) did not work.
> > > > - Regarding, the insight "Instance-wise variation is important for the design of GANs, which is also a clear difference between generative models and discriminative models" -> this does not sound as a novel and surprising insight to me. The GAN literature has seen many papers when modulation of intermediate features or their properties was justified to be important. This is a rather intuitive difference between generative and discriminative networks.
> > > > - Overall, my (subjective) evaluation of novelty is modest.
> > > >
> > > > ### Analysis
> > > >
> > > > - I am not convinced with the explanation in regards to self-attention. In self attention, the network learns which pixels of image should contribute to the values of all pixels in the next layer. This is still similar in motivation to MTM (with a very different implementation). It is true that receptive field in self-attention is *fixed*, but at the same time it is *full*, so the motivation that the network cannot learn long-range dependencies does not work. I would appreciate if the authors still have time to comment on this item.
> > > >
> > > > ### Overall
> > > >
> > > > Overall, I think this is a solid paper with extremely strong experimental results. On balance, the technical novelty and analysis provided in the paper are not as strong as a NeurIPS paper is expected to offer. Partially, this can be addressed with improved writing. I am borderline in my evaluation and would be waiting for the discussion among reviewers.

---

> > > > > ### Author Response · Authors · 2023-08-18
> > > > > **Clarifications on novelty and self-attention results**
> > > > >
> > > > > Thanks for your reply. We are glad that we have smoothed out your misunderstandings on the implementation of MTM, and we will follow your suggestions and revise the writing to avoid causing misunderstandings from other readers. We are also encouraged that you highly appreciate our experimental results, which we treat as one main contribution of this work. In the following, we will response to your remaining concerns on the novelty and the analyses of self-attention results.
> > > > >
> > > > > ### About novelty
> > > > >
> > > > > - Putting our motivation aside, we still believe that our MTM is novel even when one may consider MTM as "bringing some technique (*i.e.*, DCN) from non-GAN literature to GANs". With all due respect, we would like to point out the fact that such a migration has *never* been explored in previous arts. Therefore, one great insight of this work is that we bring the combination of deformable convolution and GANs into the public view, and confirm that such a combination could serve as a basic operation in GANs, which leads to a stronger baseline than the StyleGAN family. We believe that **it should be very important for the community** to know the effectiveness of using deformable convolution in GANs, which substantially improves the performance simply with a plug-in module and without hyper-parameter tuning. To our best knowledge, few papers have discussed the possibility of integrating DCN to GANs, let alone made the idea work in a general sense.
> > > > >
> > > > >
> > > > > ### About analyses of self-attention results
> > > > >
> > > > > - We agree with you that self-attention offers a *full* receptive field, however, it also brings with it the difficulty of optimization, especially when the feature map is with a large spatial dimension (like 32x32, 64x64, or even larger).
> > > > > - There are already some evidences in the field of visual perception (*e.g.*, Swin-Transformer [a] outperforms the vanilla self-attention [b] through a local-window implementation) and in the field of image generation (*e.g.*, STrans-G [c] achieves better synthesis performance than ViTGAN [d], where the former also adopts the local-window attention). Hence, we would like to point out that the practical performance of using self-attention may not always reach its theoretical capacity.
> > > > >
> > > > > ### About analyses of the receptive field of MTM
> > > > >
> > > > > - Recall that our motivation is to offer the generator in GANs an additional degree of freedom to modeling the data variation, instead of merely enlarging the receptive field for a long-range dependency. Indeed, after training, the deformable convolution happens to obtain a larger receptive field than the conventional convolution, but we attribute the phenomenon to "a larger receptive field could offer a larger variation". For example, a maximum receptive field of 3 can only allow the kernel to shift with 0 $\sim$ 3 offsets, but a maximum receptive field of 5 could let the kernel to shift with 0 $\sim$ 5 offsets.
> > > > >
> > > > > We hope the above analyses could help address your concerns, and we will also add the discussion in the revision.
> > > > >
> > > > > [a] Hierarchical Vision Transformer using Shifted Windows. Liu *et al.* ICCV'21.
> > > > >
> > > > > [b] An Image is Worth 16x16 Words: Transformers for Image Recognition at Scale. Dosovitskiy *et al.* ICLR'21.
> > > > >
> > > > > [c] STransGAN: An Empirical Study on Transformer in GANs. Xu *et al.* arXiv preprint.
> > > > >
> > > > > [d] ViTGAN: Training GANs with Vision Transformers. Lee *et al.* ICLR'22.

---

### Official Review · Reviewer_Sr9W · 2023-07-06

**Soundness:** 3 good
**Presentation:** 3 good
**Contribution:** 2 fair
**Rating:** 4
**Confidence:** 5

**Summary:**

In this paper, the authors equip the generator in generative adversarial networks (GANs) with a plug-and-play module, termed as modulated transformation module (MTM). This module predicts spatial offsets under the control of latent codes, based on which the convolution operation can be applied at variable locations for different instances, and hence offers the model an additional degree of freedom to handle geometry deformation. Extensive experiments suggest that this approach can be faithfully generalized to various generative tasks, including image generation, 3D-aware image synthesis, and video generation, and get compatible with state-of-the-art frameworks without any hyper-parameter tuning.

**Strengths:**

The paper is well written and easy to understand.

**Weaknesses:**

1. The proposed Spatial Temporal Latent Code Modulation is somewhat similar to deformable convolution.
2. The comparison methods in Table 1 are too old. StyleGAN2 and StyleGAN3 were published in 2020 and 2021, respectively, and need to be compared with more recent methods published in 2022 and 2023.
3. In Table 2, the authors also need to compare with more recent methods in 2023 such as [1,2].
4. The model complexity should be compared with the SOTA methods, such as training time, inference time, model parameters, etc.

[1]Xie, Jiaxin, Hao Ouyang, Jingtan Piao, Chenyang Lei, and Qifeng Chen. "High-fidelity 3D GAN Inversion by Pseudo-multi-view Optimization." In Proceedings of the IEEE/CVF Conference on Computer Vision and Pattern Recognition, pp. 321-331. 2023.
[2]Shi, Zifan, Yujun Shen, Yinghao Xu, Sida Peng, Yiyi Liao, Sheng Guo, Qifeng Chen, and Dit-Yan Yeung. "Learning 3d-aware image synthesis with unknown pose distribution." In Proceedings of the IEEE/CVF Conference on Computer Vision and Pattern Recognition, pp. 13062-13071. 2023.

**Questions:**

See Weaknesses

**Limitations:**

Yes

---

> ### Author Rebuttal · Authors · 2023-08-09
>
> **Q1. About "the proposed Spatial Temporal Latent Code Modulation is somewhat similar to deformable convolution".**
>
> About the rationale of performing convolution at variable locations, our philosophy is indeed similar to the family of spatial transformation networks, to which deformable convolution belongs. However, different from discriminative tasks, generative tasks require the model to learn data variations. That is the reason why we introduce latent code modulation into our MTM, which is able to link the sampling stochasticity to the generation process. The table below suggests that, without such a modulation, directly applying deformable convolution to GANs drastically harms the synthesis performance. To our best knowledge, we are the first to make the idea of spatial transformation work on generative models, and we also demonstrate its effectiveness and generalizability across various generation tasks, including image generation, video generation, and 3D-aware image synthesis. We hope our discovery could make MTM a basic operation in the future design of GANs.
>
> | TaiChi-256 | FID |
> | :- | :-: |
> | StyleGAN3                  |  21.36 |
> | *w/* deformale convolution | 192.65 |
> | *w/* MTM                   |  13.60 |
>
> **Q2. About "the comparison against StyleGAN2 and StyleGAN3 are too old".**
>
> Disagree. Even till now, StyleGAN2 and StyleGAN3 still serve as strong baselines for GAN-related studies, especially for the works that target architecture design. For example, the very recent text-to-image generation works [a][b] are developed from StyleGAN2 and StyleGAN3 as well. As in this work we would like to propose a general and fundamental operation for GANs, using StyleGAN2 and StyleGAN3 as our baselines is fair. Still, following your suggestion, we evaluate our MTM on GigaGAN [b], which is currently the most cutting edge algorithm in this field. The results are shown below, where we can observe that MTM even manages to improve the performance of such a powerful model. We hope that the additional results could help address your concern.
>
> | ImageNet-64 | FID |
> | :- | :-: |
> | GigaGAN  | 7.62 |
> | *w/* MTM | 6.73 |
>
> NOTE: The GigaGAN results are reproduced by ourselves due to the fact that the official implementation is not open-sourced.
>
> [a] StyleGAN-T: Unlocking the Power of GANs for Fast Large-Scale Text-to-Image Synthesis. Sauer *et al.* ICML'23.
>
> [b] Scaling up GANs for Text-to-Image Synthesis. Kang *et al.* CVPR'23.
>
> **Q3. About the comparison with more recent methods in 2023.**
>
> With all due respect, [c] solves a different task (*i.e.*, 3D GAN inversion) from this work, making it hard to compare our MTM (which aims to improve the GAN model itself) with it. The table below includes our comparison with PoF3D [d], where our approach achieves better performance.
>
> | FFHQ-256 | FID |
> | :- | :-: |
> | PoF3D [d]  | 4.99 |
> | MTM (ours) | 4.07 |
>
> [c] High-fidelity 3D GAN Inversion by Pseudo-multi-view Optimization. Xie *et al.* CVPR'23.
>
> [d] Learning 3d-aware image synthesis with unknown pose distribution. Shi *et al.* CVPR'23.
>
> **Q4. "The model complexity should be compared with the SOTA methods, such as training time, inference time, model parameters, etc."**
>
> We transcribe Tab. 4 of our manuscript below, which indicates that our MTM works as a lightweight plug-in module.
>
> | ImageNet-128 | FID | Training time | Inference time | # Param. (MB) |
> | :- | :-: | :-: | :-: | :-: |
> | StyleGAN2 | 21.14 | 1.0× | 1.0× | 27.78 |
> | *w/* MTM  | 19.16 | 1.2× | 1.0× | 28.55 |

---

> > ### Comment · Reviewer_A6mm · 2023-08-13
> >
> > Q2: I agree with the authors here. SG2/3 are time-tested architectures that many more complicated methods build upon, so studying MTM in this setting makes sense. This is a non-issue to me, especially with the inclusion of the GigaGAN results.

---

### Official Review · Reviewer_cDZK · 2023-07-07

**Soundness:** 3 good
**Presentation:** 3 good
**Contribution:** 2 fair
**Rating:** 5
**Confidence:** 4

**Summary:**

The paper proposes a modulated transformation for GANs. Specifically, they propose learning the offset of each convolutional layer by learning additional convolution layers to predict offsets where the inputs are the latent code and the current intermediate layer features. The proposed method improves the expressive power of overall GAN framework and shows notable improvements in various situations, including GAN-based image, video, and 3D-aware generation.

**Strengths:**

- The paper is generally well-written and easy to follow.
- The proposed method is simple yet effective while showing quite a strong performance in a variety of tasks.
- The analysis (e.g., which layer should use this concept, the role of this learned transformation) is interesting.

**Weaknesses:**

- The paper argues the motivation of this paper is the limitation of AdaIN in handling complex data distribution (e.g., ImageNet). But for me, the motivation seems a bit unclear as that how this learned modulated transformation can mitigate this issue. It seems like the paper claims the proposed method helps to mitigate the limitation of AdaIN that acts at every spatial location equality by learning (possibly) non-local offsets for convolutions. Then one might expect the performance gain would be dramatic in complex datasets such as ImageNet, but it seems the gain is a bit marginal here and rather larger in other datasets or situations. Thus I suspect the improvement of the performance is simply from the increased expressive power than prior GAN architecture, not directly related to AdaIN.

- As mentioned in the discussion section, I think showing whether this type of architecture can improve the performance of other types of generative models, as the proposed method itself does not modify the AdaIN itself and thus can be used in any type of convolutional networks. For instance, Can we expect the better performance of 2D UNet-based diffusion models if we replace some convolutional layers with this method?

- For 3D-aware generation, I expect the paper should show EG3D+MTM also maintains great 3D consistency, rather than just showing the improvement in  FID score.

**Questions:**

- How much memory is increased during training (and inference) with the proposed method, compared with baselines?

**Limitations:**

The paper appropriately addresses the limitation.

---

> ### Author Rebuttal · Authors · 2023-08-09
>
> **Q1. About the motivation of mitigating the limitation of AdaIN.**
>
> This should be a misunderstanding. AdaIN has become a standard operation in GANs, which helps model the cross-instance variation. This work inherits the AdaIN design and does *not* intend to improve this operation. Instead, our point is that "the conventional convolution interacts with the feature map at *fixed* locations, resulting in limited expressive power of handling *spatial* variations". We suspect that this might be the reason why GANs usually work well on datasets where all instances are with a similar shape (*e.g.*, human face), but perform poorly on datasets where instances are with large shape variations (*e.g.*, human body movements in TaiChi). This motivates us to propose MTM, which allows the convolutional kernels to act on the feature map at *variable* locations. It is noteworthy that MTM is also controlled by the latent code (similar to AdaIN) such that different samples will be convoluted at different locations. Experiments suggest that MTM indeed helps improve the performance of GANs on datasets with large shape variations, such as TaiChi (FID from 21.36 to 13.6) and LSUN churches (FID from 4.04 to 2.32). We also included some qualitative comparisons in Fig. R2 (see **the newly uploaded one-page PDF**).
>
> **Q2. About “performance gain on ImageNet is marginal.”**
>
> With all due respect, we do *not* think improving the FID of StyleGAN2 on ImageNet-128 from 21.14 to 19.16 is marginal. Considering the fact that ImageNet is with complex data distribution, improving the generative performance on ImageNet is not an easy thing, let alone we only introduce a lightweight plug-in module (*e.g.*, MTM) without any other modifications or hyper-parameter tuning. Meanwhile, StyleGAN2 only employs around 30 million parameters, which could be the major bottleneck under such a challenging setting. To verify this hypothesis, we evaluate our approach on a more capable GAN model (*i.e.*, GiganGAN [a]), and confirm that our MTM can also improve the performance of such a strong baseline (with 210 million parameters). We hope that the additional results could help address your concern.
>
> | ImageNet-64 | FID |
> | :- | :-: |
> | GigaGAN  | 7.62 |
> | *w/* MTM | 6.73 |
>
> NOTE: The GigaGAN results are reproduced by ourselves due to the fact that the official implementation is not open-sourced.
>
> [a] Scaling up GANs for Text-to-Image Synthesis. Kang *et al.* CVPR'23.
>
> **Q3. About “the improvement of the performance is simply from the increased expressive power, not directly related to AdaIN”.**
>
> You are correct that our MTM indeed offers a promising way to enhance the expressive power of generators in GANs, with only a few more parameters and computation overheads (see the table below). Meanwhile, as stated in Q1, this work does *not* target improving the AdaIN operation. In fact, AdaIN has inspired style modulation, which provides a good solution to modeling the cross-instance variation. Its core idea is to use the latent code to control the feature map modulation. However, style modulation usually works with conventional convolution, where the convolution position is shared across instances. Our key motivation is to offer the generator an additional degree of freedom (*i.e.*, the spatial convolutional locations), which is also controlled by the latent code. We are sorry for the misunderstanding we have caused, and we will revise the presentation to make our motivation clearer.
>
> | ImageNet-128 | FID | Training time | Inference time | # Param. (MB) |
> | :- | :-: | :-: | :-: | :-: |
> | StyleGAN2 | 21.14 | 1.0× | 1.0× | 27.78 |
> | *w/* MTM  | 19.16 | 1.2× | 1.0× | 28.55 |
>
> **Q4. About applying MTM to 2D UNet-based diffusion models.**
>
> Diffusion models adopt a different philosophy from GANs to model the data variation. More concretely, there is no concept of "latent code" in diffusion models. Recall that, the rationale behind MTM is to use latent code to modulate the convolutional positions of each instance. From this perspective, it is not straightforward to test MTM on diffusion models.
>
> **Q5. About the 3D consistency evaluation on the task of 3D-aware generation.**
>
> Thanks. Following EG3D, we calculate the depth error for 3D consistency evaluation. The results are listed in the table below, where a smaller depth error means better 3D consistency. We can tell that our MTM improves the synthesis performance (*i.e.*, measured by FID) without sacrificing the 3D consistency. We also include a new figure (Fig. R1) in **the newly uploaded one-page PDF** to visualize the geometry of synthesized samples from various viewing points. All these results will be included in the revision.
>
> | FFHQ-256 | FID | Depth Error |
> | :- | :-: | :-: |
> | EG3D     | 4.32 | 0.328 |
> | *w/* MTM | 4.07 | 0.336 |
>
>
> **Q6. About the training/inference memory comparison against baselines.**
>
> Following the suggestion, we list the GPU memory cost of both training and inference stages in the table below. Thanks to the lightweight design of our MTM, the memory cost barely increases.
>
> | ImageNet-128 | Training (batch size 128) | Inference (batch size 1) |
> | :- | :-: | :-: |
> | StyleGAN2 | 32.17G | 2731M |
> | *w/* MTM  | 32.34G | 2751M |

---

> > ### Comment · Reviewer_cDZK · 2023-08-13
> > **Response**
> >
> > Thanks for the detailed response. For Q1 and Q3, I now understand the point but please edit your introduction slightly if the paper is accepted to reflect your response, as two paragraphs are mentioning AdaIN in the section and some readers might misunderstand the purpose of the work.
> >
> > For Q2, I am not saying the performance gain in ImageNet is marginal; I wanted to say the improvement seems "relatively" marginal. As you mentioned, ImageNet is a complex dataset and I expect improvement should be much larger than this dataset considering the motivation; but it seems the improvements is larger in other fine-grained dataset (e.g., Taichi). That being said, considering the results with GigaGAN, I understand the point.
> >
> > For Q4, I understand the point.
> >
> > For Q5, I wonder why the depth error becomes larger. For me FID improvement and the performance drop in depth error seems compatible and I have a concern if this method really works well in 3D generation as well. At least authors should provide some analysis in this results, not just stating "without sacrificing the 3D consistency." because one can state the opposite side as well with the provided results: EG3D w/o MTM works better by achieving better 3D consistency without sacrificing FID.
> >
> > If Q5-related concerns is addressed, I will raise my score.

---

> > > ### Author Response · Authors · 2023-08-14
> > > **Response to Q5**
> > >
> > > We are glad that our previous responses have addressed most of your concerns and we will make our introduction clearer to avoid misunderstanding in the revision. Thanks again for your suggestions. In the following, we would like to make further explanations on Q5.
> > >
> > > - First, please allow us to recapitulate the **setting and instantiation** of 3D-aware image synthesis. Recall the formulation of EG3D, which first employs a 2D backbone to generate triplane features and then decodes the triplane features to a 2D image via volumetric rendering. We would like to clarify that the proposed MTM is directly applied to the 2D backbone, in the same way as how it is applied to 2D image generation (*e.g.*, TaiChi), and we inherit the rendering pipeline from EG3D.
> > >
> > > - Second, we provide **detailed analyses** of the comparison results regarding FID and depth error, which are transcribed below. (i) Learning 3D-aware image synthesis from 2D datasets is challenging and hence previous attempts usually observe a performance (*i.e.*, quality and diversity of the synthesized images) gap between 2D generators and 3D-aware generators. For example, the state-of-the-art 3D-aware model, EG3D (FID 4.32), is still left behind by the 2D model, StyleGAN2 (3.78). Our MTM helps *narrow down this gap (from 0.54 to 0.29) with only marginal computational overheads*. (ii) The reviewer is right that 3D consistency is an important metric to evaluate 3D-aware image synthesis. However, we need to recap that our module is only applied to the 2D backbone and enables 2D learnable deformation offsets, hence it could barely contribute to the learning of 3D geometry due to the lack of explicit 3D supervision or 3D modeling. Here, by saying "without sacrificing the 3D consistency", we would like to deliver the message that *we are encouraged that the introduction of our MTM does not weaken the model's capability of learning 3D geometry*.
> > >
> > > | FFHQ-256  | FID | Depth Error |
> > > | :- | :-:  | :-: |
> > > | StyleGAN2 | 3.78 | -     |
> > > | EG3D      | 4.32 | 0.328 |
> > > | *w/* MTM  | 4.07 | 0.336 |
> > >
> > > - Third, we would also like to point out the **limitation of existing approaches for 3D-aware image synthesis**, which heavily rely on a well-defined canonical space. Hence, exisiting methods are commonly evaluated on well-aligned datasets (such as human faces) and also require the ground-truth object pose of each training sample, hindering them from being applied to datasets with large shape variations. That is the reason why we fail to evaluate the compatiblity between our MTM and 3D-aware image synthesis on a more challenging dataset.
> > >
> > > - Fourth, inspired by your comments, we would like to discuss some **future work** of our MTM. For example, it is possible to introduce our MTM to both the 2D backbone and the triplane features, to further offer the generator a degree of freedom regarding 3D geometry deformation.
> > >
> > > We will include the above discussion in the revision to help readers better interpret the experimental results, as well as understanding the scope of this work. Thank you for pointing this out to us.

---

### Author Rebuttal · Authors · 2023-08-09

Thank all reviewers for their valuable comments and suggestions. We additionally included some geometry of synthesized samples from various viewing points and qualitative comparisons in **the newly uploaded one-page PDF**.

---

### Decision · Program_Chairs · 2023-09-21

**Decision:**

Accept (poster)

**Comment:**

1x A, 1x BA, and 2x BR. This paper proposes the modulated transformation modules that learn the kernel spatial offsets in CNN-based GAN generators, so as to enable increased degree of freedom and handle geometry deformation better. Most reviewers recognize its (1) clear presentation, (2) simple yet effective method, and (3) strong experimental results. The major concerns of the borderline reviewers lie in its modest novelty compared to the previous work DCN. Other concerns including the comparisons to addition baselines have been addressed by the rebuttal. The AC leans to accept the paper due to its solid implementation and empirical studies.